# COMPOSITIONAL LANGUAGE CONTINUAL LEARNING

**Yuanpeng Li,**[*] **Liang Zhao, Kenneth Church**
Baidu Research

**Mohamed Elhoseiny**[†]
KAUST, Stanford University

## ABSTRACT

Motivated by the human's ability to continually learn and gain knowledge over time, several research efforts have been pushing the limits of machines to constantly learn while alleviating catastrophic forgetting (Kirkpatrick et al., 2017b). Most of the existing methods have been focusing on continual learning of label prediction tasks, which have fixed input and output sizes. In this paper, we propose a new scenario of continual learning which handles sequence-to-sequence tasks common in language learning. We further propose an approach to use label prediction continual learning algorithm for sequence-to-sequence continual learning by leveraging compositionality (Chomsky, 1957). Experimental results show that the proposed method has significant improvement over state-of-the-art methods. It enables knowledge transfer and prevents catastrophic forgetting, resulting in more than 85% accuracy up to 100 stages, compared with less than 50% accuracy for baselines in instruction learning task. It also shows significant improvement in machine translation task. This is the first work to combine continual learning and compositionality for language learning, and we hope this work will make machines more helpful in various tasks. The code is available at: https://github.com/yli1/CLCL.

## 1 INTRODUCTION

Continual Learning is a key element of human intelligence that enables us to accumulate knowledge from a never ending stream of data. From machine learning perspective, there is no guarantee that information accessed at a current task to be revisited later in future tasks. This leads to what is known as Catastrophic Forgetting (McCloskey & Cohen, 1989; McClelland et al., 1995); significant drop in previously obtained knowledge of an AI system as it learns new information and gets less/no exposure to old information. Several approaches have been proposed to bridge the gap between machine and human continual learning skills with catastrophic forgetting being the central problem. Existing continual learning methods have focused mostly on classification tasks (e.g. (Rebuffi et al., 2017; Lopez-Paz & Ranzato, 2017; Shin et al., 2017; Li & Hoiem, 2016; Shmelkov et al., 2017; Triki et al., 2017; Li & Hoiem, 2016; Triki et al., 2017; Rusu et al., 2016; Lee et al., 2017; Elhoseiny et al., 2018; Kirkpatrick et al., 2017c; Zenke et al., 2017; Chaudhry et al., 2018)).

In this paper, we propose a new scenario of continual learning which handles sequence-to-sequence tasks common in language learning. Continual language learning is an open question which has not been studied extensively in machine learning and NLP domains. It may facilitate a variety of applications in NLP systems. For example, it enables a robot to keep on learning new tasks via natural language instruction, a conversational agent to adapt to new conversation topics quickly, and a neural machine translation system to expand its vocabulary continually.

Humans learn language by leveraging *systematic compositionality*, the algebraic capacity to understand and produce large amount of novel combinations from known components (Chomsky, 1957; Montague, 1970). Compositional generalization is critical in human cognition (Minsky, 1986; Lake et al., 2017). It also helps humans acquire language from a small amount of data, and expand vocabulary sequentially (Biemiller, 2001). In contrast to humans' such ability, state-of-the-art continual learning approaches do not achieve the expected generalization. Table 1 and Figure 3 show the performance of state-of-the-art approaches (Kirkpatrick et al., 2017a; Aljundi et al., 2018) when tested

---

[*]Corresponding author: yuanpeng16@gmail.com
[†]Work partially done while visiting Baidu Research

in instruction learning and machine translation tasks. This highlights the lack of generalization of these approaches, designed after classification tasks, on sequence generation language tasks and the importance of studying the design of continual learning methods for language learning.

| Method \Stage | Transfer | | | Forget | | | Long-forget | | |
|---|---|---|---|---|---|---|---|---|---|
| | 1 | 10 | 100 | 1 | 10 | 100 | 1 | 10 | 100 |
| Standard | 2.3 | 0.2 | 0.0 | 30.8 | 0.9 | 0.0 | 30.8 | 11.2 | 7.9 |
| Compositional | 98.8 | 15.0 | 0.0 | 99.3 | 71.7 | 0.7 | 99.3 | 85.5 | 47.4 |
| EWC | 2.8 | 0.2 | 0.0 | 35.0 | 1.0 | 0.2 | 35.0 | 11.5 | 11.1 |
| MAS | 0.6 | 0.2 | 0.0 | 20.0 | 0.8 | 0.1 | 20.0 | 10.8 | 9.8 |
| Proposed | **99.9** | **99.8** | **90.7** | **100.0** | **99.9** | **89.5** | **100.0** | **100.0** | **86.0** |

Table 1: Mean of evaluation accuracy (%) on instruction learning tasks (Section 4 for details). Baselines include Compositional (Li et al., 2019), EWC (Kirkpatrick et al., 2017a), and MAS (Aljundi et al., 2018). Please refer to Table 3 in Appendix for more results and standard deviations.

Modeling continual language learning with improved compositional understanding is at the heart of this paper. More concretely, we address the challenge of open and growing vocabulary problem with continual learning. It requires optimizing over two objectives. First, previously learned knowledge should be transferred and combined with new knowledge. Second, the learned model should resist catastrophic forgetting (Kirkpatrick et al., 2017b), where a model adapted to a new distribution no longer works on the original one. To achieve these objectives, we use compositionality to separate semantics and syntax of an input sentence, so that we can convert label prediction algorithm to sequence to sequence algorithm for continual learning.

The contributions of this paper can be summarized as follows.

- We propose a new scenario of continual learning which handles sequence-to-sequence tasks common in language learning.

- We propose an approach to use label prediction continual learning algorithm for sequence-to-sequence continual learning by leveraging compositionality. To our knowledge, this is the first work for applying compositionality to continual learning of sequence-to-sequence tasks, targeting at both knowledge transfer to later stages and catastrophic forgetting prevention on previous stages.

- Experiments show that the proposed method has significant improvement over multiple state-of-the-art baselines in both knowledge transfer and catastrophic forgetting prevention with almost 85% accuracy up to 100 stages on language instruction task. It also shows significant improvement in a machine translation task.

## 2  RELATED WORK

Our work is closely related to compositionality and continual learning or lifelong learning. Here, we briefly review some related work in these areas.

**Compositionality**   Compositional generalization is critical in human cognition (Minsky (1986); Lake et al. (2017)), and it helps humans acquire language from a small amount of data, and expand vocabulary sequentially (Biemiller (2001)). Therefore, researchers have been studying how to enable human-level compositionality in neural networks for systematic behaviour (Wong & Wang, 2007; Brakel & Frank, 2009), counting ability (Rodriguez & Wiles, 1998; Weiss et al., 2018) and sensitivity to hierarchical structure (Linzen et al., 2016). Recently, people proposed multiple related tasks (Lake & Baroni, 2018; Loula et al., 2018; Lake et al., 2019) and methods (Lake & Baroni, 2018; Loula et al., 2018; Kliegl & Xu, 2018) with different kinds of RNN models and attention mechanisms. Though these methods enable generalization when the training and test sentences have small difference, it has been an open problem (Yang et al., 2019) to reach human-level compositionality generalization. More recently, Li et al. (2019) proposed an entropy regularization method that achieves high performance on several NLP tasks. In this paper, we study compositionality from continual learning angle. By leveraging the compositional learning approach, we propose the con-

tinual learning algorithm by encoding compositionality into DNN. To our knowledge, our work is the first to apply compositionality to continual learning in DNN.

**Continual learning**    Continual learning or lifelong learning involves multiple stages. Each stage has a set of classes and corresponding data, and the training can only access the data in the current stage. Based on the way for overcoming catastrophic forgetting, continual learning work may be categorized into data-based and model-based approaches. In *data-based approaches*, some methods store previous data either with replay buffer (Rebuffi et al., 2017; Lopez-Paz & Ranzato, 2017) or generative model (Shin et al., 2017); other approaches (Li & Hoiem, 2016; Shmelkov et al., 2017; Triki et al., 2017), employ the new task data to estimate and preserve the model behavior on previous tasks, mostly via a knowledge distillation loss as proposed in *Learning without Forgetting* (Li & Hoiem, 2016). These approaches are typically applied to a sequence of tasks with different output spaces. To reduce the effect of distribution difference between tasks, (Triki et al., 2017) propose to incorporate a shallow auto-encoder to further control the changes to the learned features, while (Aljundi et al., 2017) train a model for every task (an expert) and use auto-encoders to help determine the most related expert at test time given an example input. *In model-based approaches*, some methods dynamically increase model size for the growing information (Rusu et al., 2016; Xu & Zhu, 2018); other methods (Fernando et al., 2017; Lee et al., 2017; Kirkpatrick et al., 2017c; Zenke et al., 2017) focus on the parameters of the network. The key idea is to define an importance weight $\omega_k$ for each parameter $\theta_k$ in the network indicating the importance of this parameter to the previous tasks. When training a new task, network parameters with high importance are discouraged from being changed. In *Elastic Weight Consolidation*, (Kirkpatrick et al., 2017c) estimate the importance weights $\Omega$ based on the inverse of the Fisher Information matrix. (Zenke et al., 2017) propose *Synaptic Intelligence*, an online continual model where $\Omega$ is defined by the contribution of each parameter to the change in the loss, and weights are accumulated for each parameter during training. *Memory Aware Synapses* (Aljundi et al., 2018) measures $\Omega$ by the effect of a change in the parameter to the function learned by the network, rather than to the loss. This allows to estimate the importance weights not only in an online fashion but also without the need for labels. Finally, *Incremental Moment Matching* (Lee et al., 2017) is a scheme to merge models trained for different tasks. Model-based methods seem particularly well suited for our setup, given that we work with an embedding instead of disjoint output spaces. In this paper, we propose a method with minimal increase of model structure in each stage, and we leverage compositionality with explainable mechanisms that align with human learning.

## 3    CONTINUAL LEARNING WITH COMPOSITIONALITY

### 3.1    PROBLEM DEFINITION

Conventional continual learning algorithms are designed after fixed size input and label classification output. However in many tasks, such as language, both input and output are sequences and bridging this gap between continual learning and sequence-to-sequence models is at the heart of our work. We facilitate more accurate continual sequence-to-sequence artificial learner by proposing an approach that can leverage Label Prediction Continual Learning (*LP-CL*) compositionally into Sequence-to-Sequence Continual Learning (*S2S-CL*) .

**LP-CL: Label Prediction Continual Learning**    In LP-CL, we consider a word to label mapping problem, with input word $x$ and corresponding output label $y$. In initial learning stage, $y$ takes one of $K$ classes: $y \in V_{\text{init}} = \{c_1, c_2, \ldots, c_K\}$. In continual learning stage, $y$ takes a new class: $y \in V_{\text{cont}} = \{c_{K+1}\}$. In test, $y$ takes all previous classes: $y \in V_{\text{init}} \cup V_{\text{cont}}$. For example, in language instruction task, input $x$ is a primitive word, and output $Y$ is the corresponding action symbol; in word-level machine translation, input $x$ is an English content word, and output $Y$ is the corresponding French word. In initial training stage, we have multiple input word and output symbol pairs. In continual learning stage, we have a new input and output word pair. We train a model in initial training stage, and do not use the initial training data any longer in the rest of training stages. We then switch to the data in continual learning stage, and continually update the model. In test stage, we evaluate whether model can predict labels from both initial and continual learning stages. We denote label prediction continual learning model (LP-CL) as $P(y|x; \theta)$ .

**S2S-CL: Sequence to Sequence Continual Learning**   For sequence to sequence continual learning (S2S-CL), we consider sequential input $X = x_1, x_2, \ldots, x_n$ and output $Y = y_1, y_2, \ldots, y_m$. Each output label $y_i, i \in \{1, \ldots, m\}$ is from the corresponding label set in label prediction problem. We want to make a model $P(Y|X)$ for sequence to sequence continual learning.

*Our goal* is to facilitate better Sequence to Sequence Continual Learning (S2S-CL) capability quantified as $P(Y|X)$ by leveraging access and joint-learning with Label Prediction Continual Learning (LP-CL) model, $P(y|x; \theta)$.

### 3.2   USE LP-CL ALGORITHM FOR S2S-CL WITH COMPOSITIONALITY

The core idea of this work is to use compositionality to separate semantics and syntax, so that we can convert label prediction algorithm to sequence to sequence algorithm for continual learning. In Kirkpatrick et al. (2017a), continual learning can be probabilistically defined as follows.

$$\log P(\psi|\mathcal{D}) = \log P(\mathcal{D}_T|\psi) + \log P(\psi|\mathcal{D}_{1\ldots T-1}) - \log P(\mathcal{D}_T)$$

Here, $\log P(\mathcal{D}_T|\psi)$ is the negative of loss function in task $T$, and $\log P(\psi|\mathcal{D}_{1\ldots T-1})$ is regularization related to parameters learned during tasks $1 \cdots T - 1$. In this work, we have two parts of parameters $\psi = \theta, \phi$ for semantics $\theta$ and syntax $\phi$ processing. With compositionality (Li et al., 2019), we make $\theta$ and $\phi$ conditionally independent given $\mathcal{D}_{1\ldots T-1}$.

$$\log P(\psi|\mathcal{D}) = \log P(\mathcal{D}_T|\psi) + \log P(\theta, \phi|\mathcal{D}_{1\ldots T-1}) - \log P(\mathcal{D}_T)$$
$$= \log P(\mathcal{D}_T|\psi) + \log P(\theta|\mathcal{D}_{1\ldots T-1}) + \log P(\phi|\mathcal{D}_{1\ldots T-1}) - \log P(\mathcal{D}_T)$$

We assume syntax $\phi$ does not change after the initial stage, so we realize regularization $\log P(\phi|\mathcal{D}_{1\ldots T-1})$ by freezing $\phi$ during learning in task $T$. We use label prediction continual learning algorithm for regularization $\log P(\theta|\mathcal{D}_{1\ldots T-1})$. Please see Figure 1 for illustration.

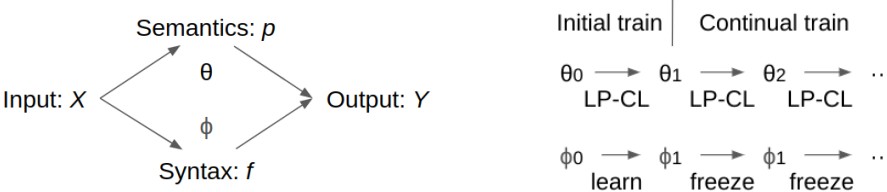

Figure 1: Flowcharts. We use compositionality to separate semantics and syntax (left). We use label prediction continual learning algorithm for $\theta$ (semantics), and freeze $\phi$ (syntax) during continual train (right).

We derive the proposed approach based on the above ideas. To use label prediction algorithm in sequence to sequence problem, we need to extract label prediction problem from sequence to sequence problem. Language is generally composed of semantics $p$ and syntax $f$, so that we decompose an input sequence to them with compositionality.

In language instruction task, for example, input $X$ is a word sequence, and output $Y$ is a label sequence. We consider $X$ has two types of information: which labels are present ($X^p$), and how the labels should be ordered ($X^f$). $Y$ is constructed by the output label types ($Y^p$), and output label order ($Y^f$) (Eq. 1). We then use chain rule (Eq. 2). With compositionality, $Y^f$ functionally depends only on $X^f$, and given $Y^f$, $Y^p$ depends only on $X^p$ (Eq. 3). For intuitive example, in language instruction, order of output actions depends only on input function words (syntax), and given the order, each output action (semantics) depends only on the corresponding input primitive. In machine translation, output order depends only on input part-of-speech information (syntax), and given the order, each output word label (semantics) depends only on the corresponding input word.

$$P(Y|X) = P(Y^f, Y^p|X^f, X^p) \tag{1}$$
$$= P(Y^f|X^f, X^p)P(Y^p|Y^f, X^f, X^p) \tag{2}$$
$$= P(Y^f|X^f)P(Y^p|Y^f, X^p) \tag{3}$$

We use LP continual learning for S2S continual learning (our goal) by decomposing output sequence to labels. We assume that the labels $y_1^p, \ldots, y_m^p$ are conditionally independent given output syntax $Y^f$ and input semantic information $X^p$ (Eq. 4) (we design model in this way). We then use total probability (Eq. 5). We further design that $x_i^p$ depends only on $y_j^f$ and $X^p$ (attention mechanism), and with label prediction component, $y_j^p$ depends only on $x_i^p$ (Eq. 6).

$$P(Y|X) = P(Y^f|X^f) \prod_{j=1}^{m} P(y_j^p|Y^f, X^p) \tag{4}$$

$$= P(Y^f|X^f) \prod_{j=1}^{m} \sum_{i=1}^{n} P(x_i^p|Y^f, X^p) P(y_j^p|x_i^p, Y^f, X^p) \tag{5}$$

$$= P(Y^f|X^f) \prod_{j=1}^{m} \sum_{i=1}^{n} P(x_i^p|y_j^f, X^p) P(y_j^p|x_i^p) \tag{6}$$

$P(x_i^p|y_j^f, X^p)$ is an operation to apply attention map $y_j^f$ on a sequence of value vectors $X^p$, so that it does not have parameters. Let $\theta$ be the parameter for label prediction module $P(y_j^p|x_i^p; \theta)$, and $\phi$ be the parameter for attention map generator $P(Y^f|X^f; \phi)$.

$$P(Y|X) = P(Y^f|X^f; \phi) \prod_{j=1}^{m} \sum_{i=1}^{n} P_{\text{att}}(x_i^p|y_j^f, X^p) P(y_j^p|x_i^p; \theta)$$

Since we assume continual learning stage does not contain new syntax patterns, we can freeze $\phi$ during continual learning stage. $\theta$ is the parameter for label prediction module. Therefore, we can use label prediction continual learning model (LP-CL) to enable compositional sequence to sequence continual learning (S2S-CL) as we detail in the next subsection.

## 3.3 DISENTANGLE SEMANTIC AND SYNTACTIC REPRESENTATIONS

Our S2S-CL approach is inspired by the idea of decomposing syntactic and semantic representations with compositionality (Li et al., 2019). Note that it is not a continual learning approach but shows how compositionality can be encoded in sequence-to-sequence models. The method disentangles syntactic and semantic[1] representations by using two representations. One generates attention maps, and the other maps attended word to action. It reduces entropy of the representations.

Suppose there are input $x$ and output $y$. $x$ contains a sequence of words, where each input word is from an input vocabulary of size $U$. $y$ contains a sequence of output symbols, where each output symbol is from an output vocabulary of size $V$. Both vocabularies contain an end-of-sentence symbol which appears at the end of $x$ and $y$, respectively. The model output $\hat{y}$ is a prediction for $y$. Suppose both input words $x_1, \ldots, x_n$ and output symbols $y_1, \ldots, y_m$ are in one-hot representation.

$$x = [x_1, \ldots, x_n], \qquad\qquad y = [y_1, \ldots, y_m]$$

To disentangle information, an input sentence $x$ is converted to semantic representation $p$ and syntactic representation $f$. Specifically, each word is encoded with two embeddings.

$$p_i = \text{Emb}_p(x_i) \in \mathbb{R}^{k_p}, \qquad\qquad f_i = \text{Emb}_f(x_i) \in \mathbb{R}^{k_f}$$

Then, they are concatenated to form two representations for the entire input sequence, i.e.,

$$p = [p_1, \ldots, p_n] \in \mathbb{R}^{k_p \times n}, \qquad\qquad f = [f_1, \ldots, f_n] \in \mathbb{R}^{k_f \times n}$$

Entropy regularization is introduced to achieve disentanglement by regularizing the $L_2$ norm of the representations $\mathcal{L}_{\text{regularize}} = L_2(p) + L_2(f)$, and then adding noise to the representations.

$$p' = p + \alpha\epsilon_p \in \mathbb{R}^{k_p \times n}, \epsilon_p \sim \mathcal{N}(0, I), \qquad f' = f + \alpha\epsilon_f \in \mathbb{R}^{k_f \times n}, \epsilon_f \sim \mathcal{N}(0, I)$$

$f'$ is fed to a sequence-to-sequence module for decoding. At each step $j$, the decoder generates $b_j \in \mathbb{R}^n$, and attention map $a_j$ is obtained with Softmax. With the attention map, weighted average

---

[1] We make a loose usage of syntactic and semantic. In natural instruction learning, syntactic refers to functional and semantic refers to primitive.

$v_j$ on noised semantic representations $p'$ is computed. Then it is fed to a fully connected one-layer network $f_{\text{predict}}$ to get score $l_j$, and a Softmax is used to compute the output distribution $\hat{y}_j$. The decoding ends if $\arg\max \hat{y}_j$ is an end-of-sentence symbol.

$$a_j = \text{Softmax}(b_j), \qquad v_j = p'a_j \in \mathbb{R}^{k_P}, \qquad l_j = f_{\text{predict}}(v_j) \in \mathbb{R}^V, \qquad \hat{y}_j = \text{Softmax}(l_j)$$

The cross entropy of $y$ and $\hat{y}$ is used as prediction loss $\mathcal{L}_{\text{predict}}$, and the final loss $\mathcal{L}$ is the combination of prediction loss and entropy regularization loss. $\lambda$ is regularization weight.

$$\mathcal{L}_{\text{predict}} = \sum_{j=1}^{m} \text{CrossEntropy}(y_j, \hat{y}_j), \qquad\qquad \mathcal{L} = \mathcal{L}_{\text{predict}} + \lambda \mathcal{L}_{\text{regularize}}$$

### 3.4 LABEL PREDICTION ALGORITHM FOR CONTINUAL LANGUAGE LEARNING

For language problems, it is natural to use non-parametric algorithm as label prediction continual learning algorithm, because input words and output actions are usually associated with embeddings. In each stage, since the original method uses two embeddings $E_r \in \mathbb{R}^{k_r}$ ($r \in \{p, f\}$) for a word and one embedding $W$ for an action, we append the new semantic $e_p$, syntactic $e_f$ and action $w$ embeddings (Figure 2). We freeze the old embedding parameters and only learn the newly added ones in the stage.

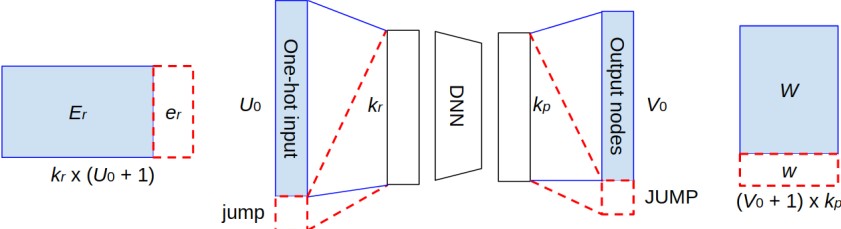

Figure 2: Illustration for the first continual learning stage. $U_0$ and $V_0$ are initial vocabulary sizes for input and output, respectively. Left is input word embedding (we only show one of two input word embeddings for simplicity). Middle is model architecture. Right is output action embedding. Parameters and data for the input word and output action embeddings of previous stage are in blue (filled boxes, solid lines), and for the new stage are in red (unfilled boxes, dashed lines). Other parts of the network are in black (unfilled boxes, solid lines).

## 4 EXPERIMENTS

We evaluate the proposed method in a continual learning task with multiple stages. The first stage is a standard process in which we train a model with combinations of multiple words in various sentence structures. In each continual stage, we add a new input word and corresponding new output symbol. The training dataset contains only one sample, whose input is a sentence with the new word, and output is a sequence with the new symbol. For each stage, we can only use the data in that stage, and have no access to data in previous or future stages.

We have two objectives in continual learning. We want previously learned knowledge to be transferred and combined with new knowledge (transfer learning), and an updated model to work on previous data (catastrophic forgetting prevention). We evaluate transfer learning by testing whether the model works on data where the new word appears with old ones (**Transfer**). We evaluate catastrophic forgetting prevention by testing whether the model works on data that only contain words up to the last stage (**Forget**). We are also interested in preventing long-term catastrophic forgetting, because it is more difficult than preventing short-term one. Thus, we test whether the new model works on the evaluation dataset in the initial stage (**Long-forget**).

**Baselines.** We designed baseline methods for compositionality Sequence-to-Sequence continual learning to validate our approach since, to our knowledge, this is the first work for continual learning of natural language instructions and machine translation. We applied standard sequence-to-sequence model (**Standard**) to our continual setting, and also the compositional generalization

method (**Compositional**) (Li et al., 2019). We also compare with state-of-the-art continual learning baselines. To fit in the experimental setting, we focus on those that do not use replay buffer, and require minimum model structure extension, so that we added **EWC** (Kirkpatrick et al., 2017a) and **MAS** (Aljundi et al., 2018) as comparable baselines due to their popularity and competitive performance in label prediction setting. The detailed implementation of the baseline and proposed methods can be found in Appendix B.

**Metric.**    We use accuracy as metric for both instruction learning and machine translation experiments. A prediction is correct if and only if it is completely identical to the ground truth. We run all experiments for five times with different random seeds.

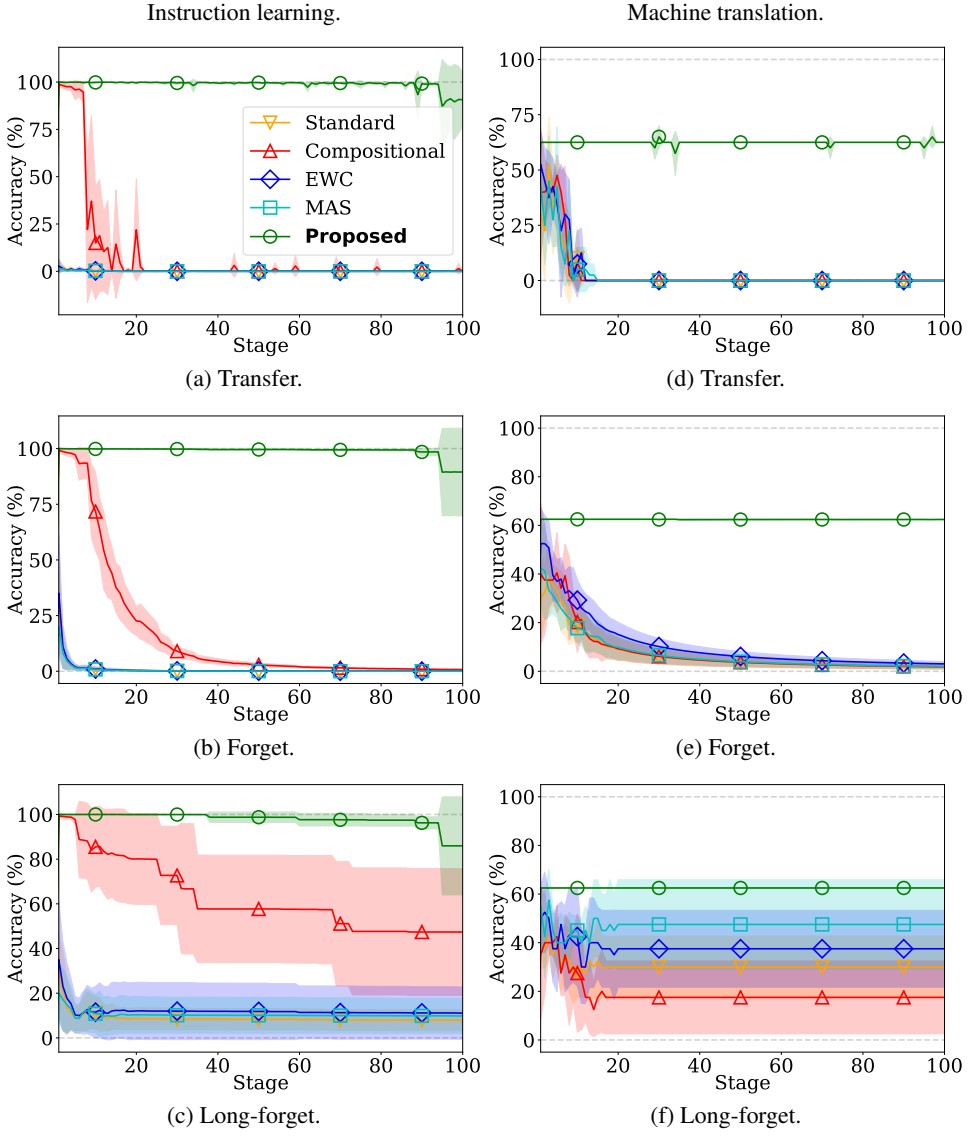

Figure 3: Mean of evaluation accuracy (%) for all methods (best viewed in color). Baselines include Compositional (Li et al., 2019), EWC (Kirkpatrick et al., 2017a), and MAS (Aljundi et al., 2018). The proposed method is significantly better than all baselines. Please refer to Figure 3 and Figure 4 in Appendix for details.

**Instruction Learning**    We first experiment on instruction learning task using SCAN dataset (Lake & Baroni, 2017). The task is summarized in Table 2 in Appendix. The details of dataset generation is in Appendix A. The results are in Figure 3 (left) and Table 1 (more details in Table 3 in Ap-

pendix). The proposed method has significantly better results than the baselines. It maintains high accuracy up to 100 stages for both transferring knowledge from previous stages to future stages, and catastrophic forgetting prevention. On the other hand, baseline methods drop performance over time. Methods without compositionality (EWC, MAS) reduces quickly, maybe because they are not designed for transferring knowledge, and since the representations are entangled, all parameters are quickly changed, causing catastrophic forgetting. Compositional method is better, but still drops, maybe because the parameters for syntax are changed over time. This experiment shows the advantage of the proposed method over baselines.

**Machine Translation** We also investigated whether the proposed approach works for other continual language learning problems. As an example, we conduct a proof-of-concept experiment for machine translation. We modified the English-French translation task in (Lake & Baroni, 2018). In each continual learning stage, we add an additional English-French word pair, in the format ("I am ENGLISH", "je suis FRENCH"). Neither English word nor French word appears in previous stages. This pair is used as training data in the stage, but test data contains other patterns. Appendix A provides more details on dataset and model configuration. The result is shown in Figure 3 (right) and Table 4 in Appendix. It shows that the proposed approach has stable and significantly higher performance than baselines. For Transfer and Forget evaluation, the baseline methods drop quickly. However, for Long-forget evaluation, they keep positive accuracy over time. This means the baseline methods have the ability to learn knowledge and remember for long time, but they are not as strong as the proposed method. This experiment shows that the proposed approach has promise to be applied to real-world tasks.

## 5 DISCUSSIONS

### 5.1 ATTENTION MAP VISUALIZATION

We hope to use compositionality for continual learning, so we want to find whether the model works in the expected mechanism. We visualize activations of attention maps on the evaluation data in the first continual stage (Figure 4).

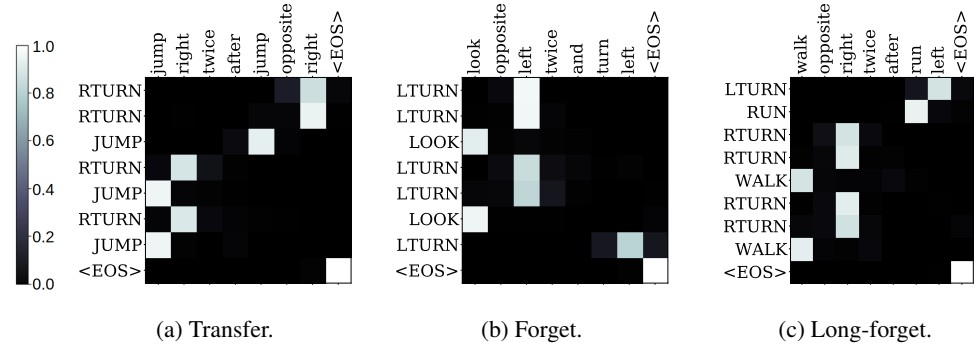

(a) Transfer.       (b) Forget.       (c) Long-forget.

Figure 4: Visualization of attention maps. The horizontal and vertical dimensions are the input and output position sequences respectively. The figures show that the model identifies the appropriate input to output position mapping. This indicates that the proposed method successfully leverages compositionality in continual learning.

The visualization shows that, for each output action, the attention is on the corresponding input word. Also, for the output end-of-sentence symbol, it is on the input end-of-sentence symbol. It is consistent with the original work, and the way humans apply compositionality. This indicates that the proposed method may be applicable to other tasks where humans use compositionality.

## 5.2 Embedding Visualization

We visualize how the new embedding parameters fit in the space with predefined dimensions, and accommodate with previously learned parameters. The visualization of attention maps explains the syntactic information, and we are also interested in semantic information.

We use t-SNE (Maaten & Hinton, 2008) to project high dimensional embeddings to two dimensional space for visualization. Our analysis focuses on semantic embedding, because it reflects how new information is encoded in the model. Since action embedding shares much information with semantic embedding, and syntactic embedding is not supposed to contain new information because grammar does not change over stages, we leave them in Appendix D.

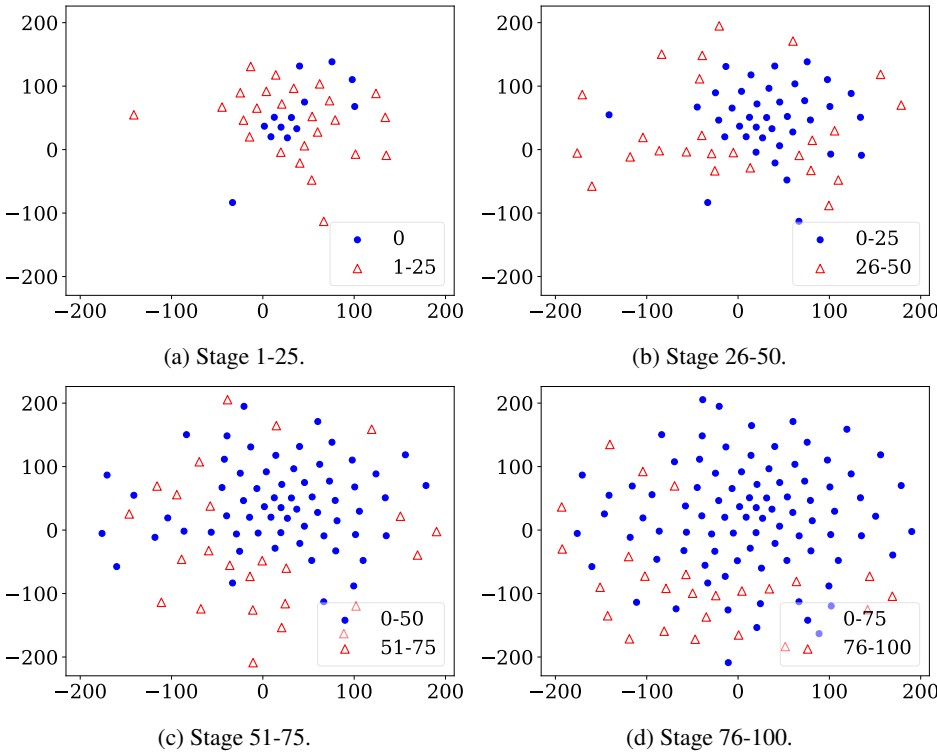

Figure 5: Embedding visualization for semantic embeddings. We see two phases. In (1-50), embeddings explore outside space. In (51-100), embeddings squeeze into the explored space.

Figure 5 shows two phases in the continual learning experiment. The first phase is from the first stage to around stage 50, where the new embeddings explore outside space. The second phase is the rest of the stages, where the embeddings squeeze into the explored space, maybe because exploring becomes expensive with the dense population under regularization. This may be an explanation for the performance decrease in the later stages of instruction learning experiment.

## 6 Conclusion

In this paper, we propose an approach to use label prediction continual learning algorithm for sequence-to-sequence continual learning problem by leveraging compositionality. To our knowledge, this is the first work to combine continual learning and compositionality for sequence-to-sequence learning. Experiments show that the proposed method has significantly better results than baseline methods, and it maintains almost more than 85% accuracy for both transfer learning and catastrophic forgetting prevention up to 100 stages. The results demonstrate that language compositionality helps continual learning of natural language instruction both efficiently and effectively. We hope this work will advance the communication between humans and machines, and make machines more helpful in various tasks.

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

## A    DATASET PREPARATION

We extend the grammar in SCAN dataset (Lake & Baroni, 2017) to generate data. For the initial stage, we generate data with three action words (walk, look, run). We then randomly and uniformly divide the data to two disjoint sets. We use one set for initial stage training data (6,601 samples), and reserve the other set as initial dataset to evaluate catastrophic forgetting in continual stages (Forget, 6,602 samples). The reserved data is also used to evaluate long-term catastrophic forgetting (Long-forget). In each continual stage, we generate data that contain a new action word (jump for the first continual stage). The data contain a one-word sentence (jump: JUMP), and we use it as training data (1 sample). We use other data in the generated data as evaluation dataset (Transfer, 7,706 samples for the first continual stage). We then add Transfer to Forget for the next stage. We set a limitation of 100,000 samples for each evaluation dataset. When it has more samples than that, we select the amount uniformly at random after generating all data once. Machine translation dataset is generated similarly from the translation dataset in SCAN dataset (Lake & Baroni, 2017). We use the original training data as the initial training data, and the original test data as the initial test data.

| Stage | Train | Transfer | Forget | Long-forget |
|---|---|---|---|---|
| Initial | walk twice | - | - | run and look |
| 1 | **jump** | **jump** twice | walk | look before run |
| 2 | **skip** | look and **skip** | look twice | walk after run |
| 3 | **crawl** | **crawl** after *skip* | *skip* after *jump* | run thrice |

Table 2: Example inputs for continual learning setup (initial and first three continual learning stages). Initial stage training data contain action words of "walk", "run" and "look". **Bold** words are introduced in current continual stage. *Italic* words are introduced in previous continual stages.

## B    IMPLEMENTATION

We use bidirectional LSTM as encoder, and unidirectional LSTM with attention as decoder for sequence to sequence architecture. The first and last states of encoder are concatenated as initial state of decoder. The state size is $h = 32$ for encoder, and $2h = 64$ for decoder. We also use $k_p = 64, k_f = 8, \alpha = 0.1$. For EWC (Kirkpatrick et al., 2017a) and MAS (Aljundi et al., 2018), we use 10 for parameter regularization weight. In initial stage, batch size is 512, and we run training 5,000 steps. In each continual stage, batch size is 1, as each continual stage only contains one sample, and we run training 1,000 steps. We have 100 continual stages. We implement all methods with TensorFlow (Abadi et al., 2016).

## C    DETAILED EXPERIMENT RESULTS

More results and standard deviation for continual learning can be found in Table 3 for instruction learning and Table 4 for machine translation experiments.

## D    EMBEDDING VISUALIZATION

Syntax embeddings in Figure 6 show that the embeddings in continual learning stages do not explore outside space, but live in the space explored in the initial stage.

Action embeddings in Figure 7 show two phases for exploration, similar with semantic embeddings. This fits expectation, because the compositionality and training encourage semantic embeddings and action embeddings to be similar.

| Method \Stage | 1 | 2 | 3 | 10 | 100 |
|---|---|---|---|---|---|
| Standard | 2.3 ± 3.0 | 1.8 ± 1.1 | 1.0 ± 1.2 | 0.2 ± 0.1 | 0.0 ± 0.0 |
| Compositional | 98.8 ± 1.5 | 98.1 ± 1.7 | 97.5 ± 1.8 | 15.0 ±29.9 | 0.0 ± 0.0 |
| EWC | 2.8 ± 3.7 | 1.4 ± 1.2 | 0.4 ± 0.3 | 0.2 ± 0.2 | 0.0 ± 0.0 |
| MAS | 0.6 ± 0.7 | 1.4 ± 1.6 | 0.3 ± 0.4 | 0.2 ± 0.1 | 0.0 ± 0.0 |
| Proposed | 99.9 ± 0.1 | 99.9 ± 0.2 | 99.8 ± 0.2 | 99.8 ± 0.3 | 90.7 ±14.5 |

(a) Transfer learning evaluation (Transfer).

| Method \Stage | 1 | 2 | 3 | 10 | 100 |
|---|---|---|---|---|---|
| Standard | 30.8 ±16.4 | 10.0 ± 8.6 | 4.3 ± 3.7 | 0.9 ± 0.6 | 0.0 ± 0.0 |
| Compositional | 99.3 ± 1.2 | 98.7 ± 1.3 | 98.2 ± 1.5 | 71.7 ±17.6 | 0.7 ± 0.2 |
| EWC | 35.0 ±25.9 | 11.0 ± 9.8 | 5.5 ± 4.9 | 1.0 ± 1.0 | 0.0 ± 0.0 |
| MAS | 20.0 ±13.3 | 8.1 ± 6.6 | 4.6 ± 4.0 | 0.8 ± 0.7 | 0.0 ± 0.0 |
| Proposed | 100.0 ± 0.0 | 100.0 ± 0.1 | 99.9 ± 0.1 | 99.9 ± 0.2 | 89.5 ±19.6 |

(b) Catastrophic forgetting evaluation (Forget).

| Method \Stage | 1 | 2 | 3 | 10 | 100 |
|---|---|---|---|---|---|
| Standard | 30.8 ±16.4 | 20.9 ±17.5 | 13.5 ±12.3 | 11.2 ± 7.2 | 7.9 ± 4.6 |
| Compositional | 99.3 ± 1.2 | 99.1 ± 1.1 | 98.9 ± 1.1 | 85.5 ±18.1 | 47.4 ±28.4 |
| EWC | 35.0 ±25.9 | 23.1 ±20.0 | 17.7 ±15.9 | 11.5 ±11.5 | 11.1 ±11.6 |
| MAS | 20.0 ±13.3 | 17.2 ±13.7 | 14.7 ±12.5 | 10.8 ± 8.5 | 9.8 ± 7.9 |
| Proposed | 100.0 ± 0.0 | 100.0 ± 0.0 | 100.0 ± 0.0 | 100.0 ± 0.0 | 86.0 ±22.0 |

(c) Long-term catastrophic forgetting evaluation (Long-forget).

Table 3: Evaluation accuracy (%) for proposed and baseline methods in instruction learning experiment. Baselines include Compositional (Li et al., 2019), EWC (Kirkpatrick et al., 2017a), and MAS (Aljundi et al., 2018).

| Method \Stage | 1 | 2 | 3 | 10 | 100 |
|---|---|---|---|---|---|
| Standard | 30.0 ±17.0 | 22.5 ±22.9 | 52.5 ±21.5 | 10.0 ±14.6 | 0.0 ± 0.0 |
| Compositional | 40.0 ±28.9 | 40.0 ±14.6 | 42.5 ±10.0 | 5.0 ±10.0 | 0.0 ± 0.0 |
| EWC | 52.5 ±14.6 | 45.0 ±15.0 | 37.5 ±20.9 | 7.5 ±15.0 | 0.0 ± 0.0 |
| MAS | 42.5 ±15.0 | 25.0 ±17.7 | 45.0 ±20.3 | 5.0 ±10.0 | 0.0 ± 0.0 |
| Proposed | 62.5 ± 0.0 | 62.5 ± 0.0 | 62.5 ± 0.0 | 62.5 ± 0.0 | 62.5 ± 0.0 |

(a) Transfer learning evaluation (Transfer).

| Method \Stage | 1 | 2 | 3 | 10 | 100 |
|---|---|---|---|---|---|
| Standard | 30.0 ±17.0 | 32.5 ±11.5 | 38.3 ±15.2 | 18.8 ± 6.8 | 1.8 ± 0.9 |
| Compositional | 40.0 ±28.9 | 37.5 ±22.4 | 37.5 ±11.5 | 20.2 ±12.6 | 1.8 ± 0.9 |
| EWC | 52.5 ±14.6 | 52.5 ±11.6 | 50.8 ±11.3 | 29.3 ± 8.5 | 3.0 ± 1.0 |
| MAS | 42.5 ±15.0 | 41.3 ±20.0 | 33.3 ±10.9 | 17.7 ± 6.0 | 2.0 ± 1.0 |
| Proposed | 62.5 ± 0.0 | 62.5 ± 0.0 | 62.5 ± 0.0 | 62.5 ± 0.0 | 62.4 ± 0.1 |

(b) Catastrophic forgetting evaluation (Forget).

| Method \Stage | 1 | 2 | 3 | 10 | 100 |
|---|---|---|---|---|---|
| Standard | 37.5 ± 7.9 | 37.5 ± 7.9 | 42.5 ±10.0 | 27.5 ± 9.4 | 30.0 ±12.7 |
| Compositional | 35.0 ±26.7 | 40.0 ±21.5 | 40.0 ± 5.0 | 27.5 ±24.2 | 17.5 ±15.0 |
| EWC | 50.0 ±13.7 | 52.5 ±16.6 | 50.0 ±11.2 | 42.5 ±20.3 | 37.5 ±15.8 |
| MAS | 55.0 ± 6.1 | 42.5 ±17.0 | 57.5 ±12.7 | 45.0 ± 6.1 | 47.5 ±18.4 |
| Proposed | 62.5 ± 0.0 | 62.5 ± 0.0 | 62.5 ± 0.0 | 62.5 ± 0.0 | 62.5 ± 0.0 |

(c) Long-term catastrophic forgetting evaluation (Long-forget).

Table 4: Evaluation accuracy (%) for proposed and baseline methods in machine translation experiment. Baselines include Compositional (Li et al., 2019), EWC (Kirkpatrick et al., 2017a), and MAS (Aljundi et al., 2018).

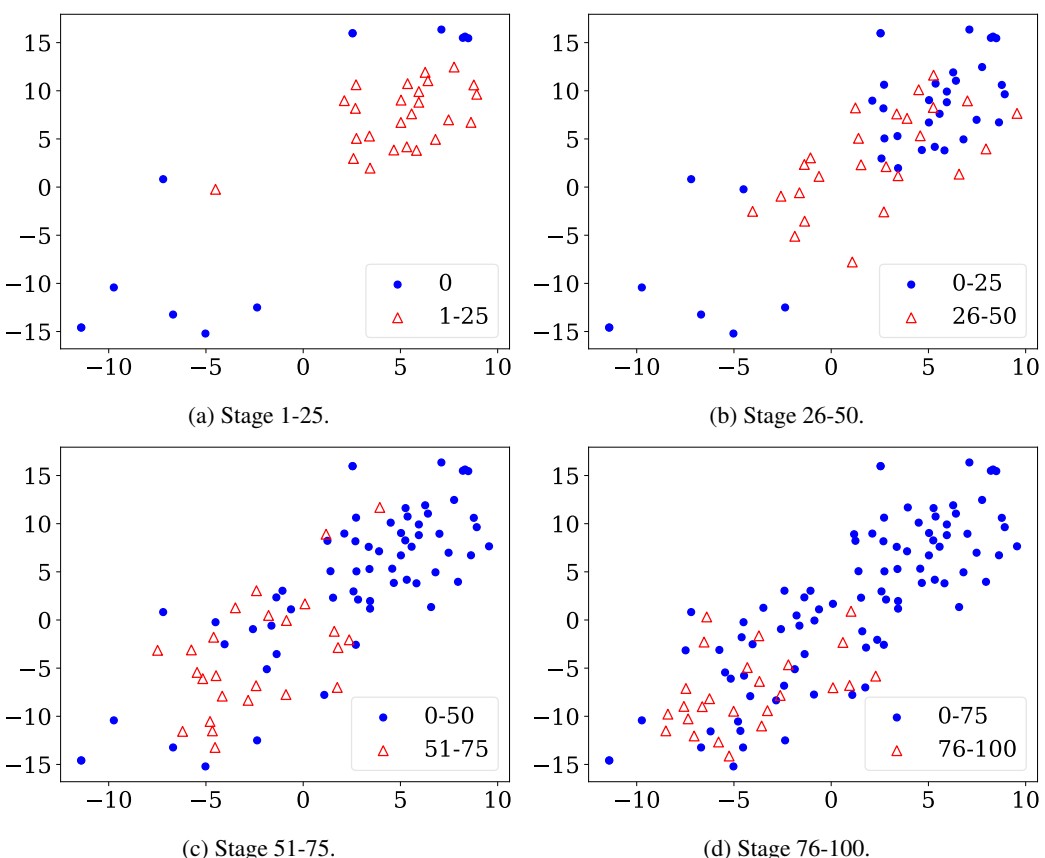

(a) Stage 1-25.

(b) Stage 26-50.

(c) Stage 51-75.

(d) Stage 76-100.

Figure 6: Embedding visualization for syntax embeddings.

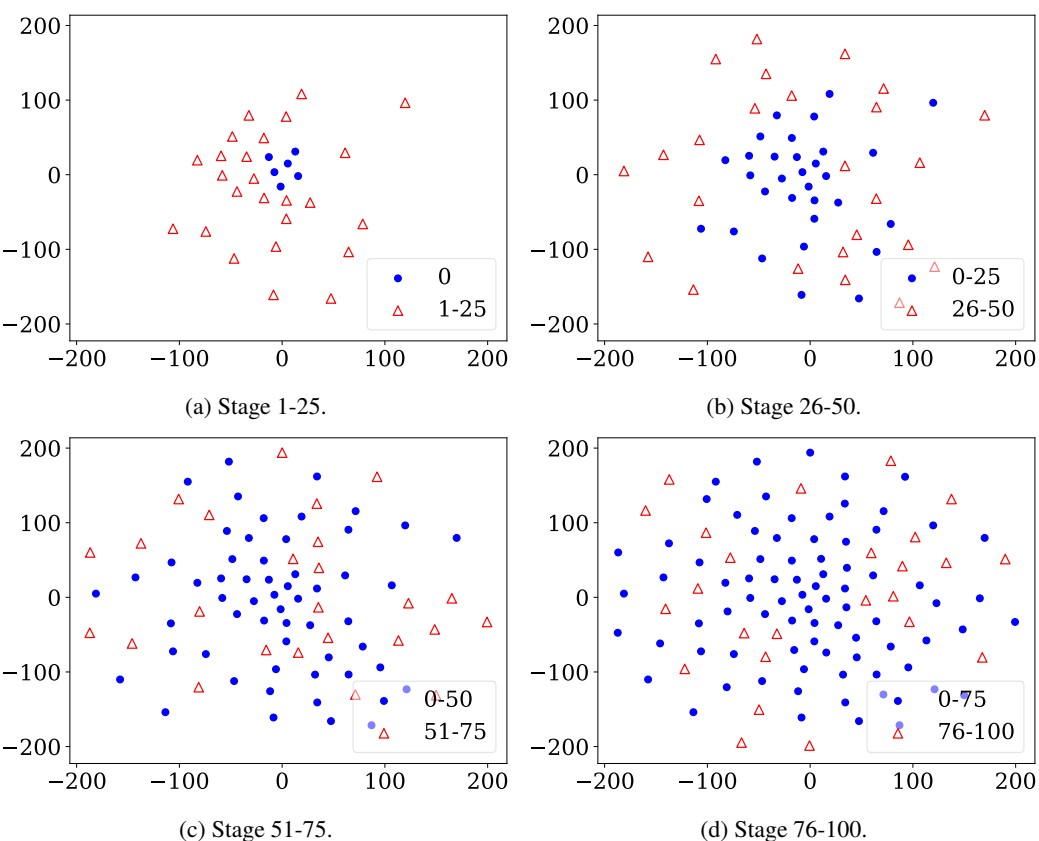

(a) Stage 1-25.

(b) Stage 26-50.

(c) Stage 51-75.

(d) Stage 76-100.

Figure 7: Embedding visualization for action embeddings.

