# OpenReview forum: "Compositional Language Continual Learning"
_ICLR.cc/2020/Conference — Accept (Poster)_

### Official Review · AnonReviewer2 · 2019-10-21
**Official Blind Review #2**

**Rating:** 6

**Review:**

This paper proposes an approach for continual learning for applications in sequence-to-sequence continual learning (S2S-CL). More specifically, the paper addresses the problem of growing vocabulary. The overall approach is intuitive yet quite effective. The method assumes a split between the syntax (f) and semantics (p) of the sequence -- in other words, each token is associated with two labels. Furthermore, the syntax is assumed to be shared across all the training sequences and the sequences that are not encountered during the initial training. A sequence model (LSTM) is used for learning the syntax f over the initial training and is then frozen for all the downstream tasks (i.e. continual learning). The sequence model predicts the correspondence between the output token and input tokens. Another network (a 1-layer  MLP) is used to predict the semantic label of the selected token in the target domain. Barring some notational confusion, I believe the method is very reasonable and should work well. They perform experiments on 2 datasets (Instruction learning & machine translation) in 3 continuous learning paradigms with varying difficulties and demonstrate that the proposed method significantly outperforms the baselines by an impressive margin.

I am leaning towards accepting this paper since I believe the proposed approach can be a promising direction for continual learning in S2S settings and the empirical results are convincing.

However, I have some concerns that might require clarification or additional experiments:
    1. The novelty of the proposed method is somewhat limited in my opinion. It seems it is a very simple adaptation of Li et al. 19 and the only addition to that work is the usage of a very simple label prediction scheme by fixing everything else. However, this might be okay since the experiments are quite compelling and the method is applied to a different setting.

    2. Notations and language in section 3.2-3.4 are hard to follow:
        - At the bottom of page 4, I believe some of the equations are not correct. For example, the first term of line 2 should be P(Y^f | X^f, X^p) rather than Y^p?
        - In the second equation of page 6, the second term should be v_j = p’ \cdot a_j
        - In Figure 1, k_r, E_r, and W are never defined (although their meaning can be inferred).
        - In section 3.4, it reads that new word embedding is appended to both semantic and syntactic embedding matrices but this doesn’t make sense because the syntactic network is already fixed so it shouldn’t be able to handle new symbols; therefore, I believe only new row is added to the semantic embedding.

    3. I believe that f_predict here is parameterized by \theta and \theta is also frozen during the continual learning phase which contradicts the claim at the end of section 3.2 that only \phi is frozen. Otherwise, it’s hard to see why f_predict does not suffer from catastrophic forgetting. From the provided source code, it seems that it is indeed the case that only the new embedding is being updated. In other words, the only thing happening at this stage is that the newly added embedding are optimized to adapt to the frozen f_predict.

    4. Due to point 3, I have some doubts about how scalable this approach is if the other embedding is not allowed to co-adapt. However, perhaps this problem could be alleviated if f_predict has extremely high capacity at initialization? More on this in point 6.

    5. Assuming all my assessment above is correct, it seems that the performance of the proposed method should *not* decrease at all so I would like to see more analysis on the decreasing performance in the instruction learning task (Figure 2, left column). Is this be due to the fact that f_predict or the other embedding is not allowed to update and the newly added embedding is mapped close to existing embedding? In that case, can increasing the model capacity solve this problem? I understand the paper makes argument about regularization but I believe this warrants thorough study for gauging the significance of this approach in more realistic settings.

    6. I would like to see a discussion of the short-comings of the approach and possible ways to overcome them. For example, freezing the syntactic network seems limiting in machine translation settings if a new language, say Italian, is added. Intuitively, knowing how to translate English-French should help translating English to Italian but fixing the syntax prevents this. Another example is that prior knowledge about the syntax needs to be known about the language for labeling the f and p and this can be expensive and cannot handle words that have more than 1 usages (e.g. run can be used as a noun but also as a verb).

I am willing to increase my score to accept if the revised manuscript can address the majority of, if not all of the concerns listed above.

========================================================================================================
Minor comments that did not affect my decision:
    - These sections could greatly benefit from an overall flow chart of how everything fits together
    - It says the experiments are repeated with 5 different random seeds so why not add error bars to the plots?
    - What characteristics of the 2 tasks cause the baselines to behave so differently?


**Experience Assessment:**

I have read many papers in this area.

**Review Assessment: Checking Correctness Of Derivations And Theory:**

I carefully checked the derivations and theory.

**Review Assessment: Checking Correctness Of Experiments:**

I assessed the sensibility of the experiments.

**Review Assessment: Thoroughness In Paper Reading:**

I read the paper thoroughly.

---

> ### Author Response · Authors · 2019-11-15
> **Response to Review #2**
>
> Thanks very much for your constructive comments and support.
>
> Q1: The novelty is somewhat limited. However, this might be okay since the experiments are quite compelling and the method is applied to a different setting.
> A1: Thank you. We agree that this is okay. Our novelty is to propose continual language learning, and address it with compositionality to bridge LP-CL and S2S-CL.
>
> Q2: Notations and language in section 3.2-3.4.
> A2: Thank you very much. We revised the paper and fixed notations and language.
> For the syntax embedding of a new word, the new word is still supposed to have syntactic information, but it is seen information, so that the model should still learn it to be close to some existing syntax embeddings. These syntax embeddings should correspond to empty syntactic information in instruction learning task. It might also be feasible to encode this to model design, but the current design is simpler.
>
> Q3: Freezing \theta contradicts the claim that only \phi is frozen.
> A3: In Section 3.2, we freeze \phi to keep syntax processing ability in continual learning, and leave learning semantic parameters to LP-CL algorithm. In this case, the non-parametric LP-CL algorithm freezes a part of \theta. We now made it more clear in Section 3.2.
>
> Q4: The scalability without co-adapting the other embedding. Perhaps it could be alleviated if f_predict has extremely high capacity at initialization?
> A4: This is the initial research for continual language learning, and we assume there is no new syntax information during continual learning. This assumption is reasonable to some extent, because syntax has less variations than semantics, and syntax does not change frequently. Since semantics and syntax are separated, increasing the capacity of f_predict at initialization may not address syntax continual learning, or at least it is not an efficient approach.
>
> Q5: Analysis on the decreasing performance in the instruction learning task.
> A5:  The performance may decrease during the end of the second phrase (discussion Section 5.2) of continual learning where the embeddings squeeze into the explored space, maybe because exploring becomes expensive with the dense population under regularization. We wrote the analysis more clearly in the discussion Section 5.2.
>
> Q6: Short-comings of the approach and how to overcome.
> A6: One short-coming is that this work focuses on continual learning for semantics, but not for syntax. To make it possible, we may need to address hierarchical compositionality, maybe with stacked attention models such as transformer.
>
> Q7: Overall flow chart.
> A7: We added an overall flow chart in Figure 1.
>
> Q8: Error bars to the plots.
> A8: We added standard deviations in Figure 3.
>
> Q9: What characteristics of the 2 tasks cause the baselines to behave so differently?
> A9: The 2 tasks are compositional continual language learning problems, with increasing numbers of vocabulary. The baselines do not handle such characteristics, while the proposed method does, so that the proposed method outperforms the baselines.

---

### Official Review · AnonReviewer3 · 2019-10-23
**Official Blind Review #3**

**Rating:** 8

**Review:**

*Summary

The paper proposes a continual learning algorithm for label prediction to deal with sequence-to-sequence continual learning problems. The proposed method is designed to leverage compositionality.  The key idea of the proposed method is to enable the network to represent syntactic and semantic knowledge separately. This allows the neural network to leverage compositionality for knowledge transfer while alleviating catastrophic forgetting.
The experiments showed that their method performed significantly better results than baseline methods.  The method was tested on two different datasets, e.g., instruction Learning and machine translation.


*Decision and supporting arguments

I think this paper has enough quality to be be accepted as a conference paper.
The main reasons of my decision are two-folds.
First, the proposal is quite insightful. The separation of semantics and syntax of an input sentence for using compositionality is an excellent idea.
Second, the proposed method improved the performance on two dataset significantly. This supports the usefulness of the idea.


*Additional feedback

My concern is about evaluation. Table 1 shows the significant difference between the proposed method and the baseline methods. It looks to nice. But, this suggests that the datasets might be too artificial for this evaluation. To my understanding, both of the datasets are artificial to some extent. Hopefully, the method should be evaluated on the more realistic dataset.


**Experience Assessment:**

I have read many papers in this area.

**Review Assessment: Checking Correctness Of Derivations And Theory:**

N/A

**Review Assessment: Checking Correctness Of Experiments:**

I assessed the sensibility of the experiments.

**Review Assessment: Thoroughness In Paper Reading:**

I read the paper at least twice and used my best judgement in assessing the paper.

---

> ### Author Response · Authors · 2019-11-15
> **Response to Review #3**
>
> Thanks very much for your helpful comments and support.
>
> Q1: Evaluation on the more realistic dataset.
> A1:  Yes, we agree. This is the initial research for compositional continual language learning, aiming at finding fundamental mechanism for such problem, so we started with these instruction learning and machine translation datasets. Also, artificial data helps to clearly show that the core idea is valid.

---

### Official Review · AnonReviewer1 · 2019-10-23
**Official Blind Review #1**

**Rating:** 3

**Review:**

The paper is about continual learning on NLP applications like natural language instruction learning or machine translation. The authors propose to exploit "compositionality" to separate semantics and syntax so as to facilitate the problem of interest.

In summary, the current manuscript is clearly not ready for publication. The writing is not good, as I cannot see clearly the backbone of the paper. Honestly, I got very confused by the presented contents. What’s the problem/motivation? No preliminary background knowledge? What’s the classic or naïve method to solve the problem of interest? What’s the main advantage/novelty of the presented method compared to that classic/naïve method?

Please see the detailed comments below.

In the Abstract, you mentioned "One of the key skills … ability to produce novel composition’’. Do you imply your method can continually learn new compositions? If so, how does that reflect in the technical parts and the experiments?

The paragraph before Section 3 might be redundant.

Many typos exist. Such as the word "iuput’’ in the 1st paragraph of Page 4.

What’s the problem of S2S-CL? Increasing input number n and output number m?

What does the word "COMPOSITIONALITY’’ mean in Section 3.2? Also, what’s the relationship between the last two equations of Page 4?

How do you defend the simplifications adopted in the first Equation of Page 5?

The notation "{0,1}^{Uxn}’’ usually represents a binary matrix of size Uxn. It is not suitable to use them to represent a matrix containing one-hot columns.

At the beginning of Page 6. Why entropy regularization can be introduced via L2 norm on the embedding matrixes p and f? Also why that L2 norm regularizations can `achieve disentanglement’? Please provide the detailed proof or the reference.

What are the detailed settings of the demonstrated experiments?


**Experience Assessment:**

I do not know much about this area.

**Review Assessment: Checking Correctness Of Derivations And Theory:**

I assessed the sensibility of the derivations and theory.

**Review Assessment: Checking Correctness Of Experiments:**

I assessed the sensibility of the experiments.

**Review Assessment: Thoroughness In Paper Reading:**

I read the paper at least twice and used my best judgement in assessing the paper.

---

> ### Author Response · Authors · 2019-11-15
> **Response to Review #1**
>
> Thank you for the review, question and suggestions.
>
> Q1: Writing and backbone of the paper.
> A1: We revised the paper and made the main points more clear. In this paper, we propose a new scenario of continual learning which handles sequence-to-sequence tasks common in language learning. We further propose an approach to use label prediction continual learning algorithm for sequence-to-sequence continual learning by leveraging compositionality.
>
> Q2: Problem/motivation, preliminary background knowledge
> A2: The main problem is how to enable continual language learning with compositionality. The backgrounds are covered in introduction and related work sections (there are also pointers to references).
>
> Q3: classic or naïve methods?
> A3: The classic or naïve continual learning algorithms (Kirkpatrick et al., 2017a; Aljundi et al., 2018) mostly focus on label prediction tasks (with fixed input and output sizes), but we address sequence to sequence tasks (with unfixed input and output sizes), common in language learning. We use these methods as experiment baselines.
>
> Q4: Main advantage/novelty compared to classic/naïve method?
> A4: Our approach bridges the gap between label prediction and sequence-to-sequence learning by using compositionality in language. To our knowledge, this is the first work for applying compositionality to continual learning of sequence-to-sequence tasks. Experiments show that the proposed method has significant improvement over multiple state-of-the-art baselines.
>
> Q5: Does "One of the key skills … ability to produce novel composition’’ in abstract imply your method can continually learn new compositions? How does that reflect in the technical parts and the experiments?
> A5: Yes, this skill is the compositional generalization skill, which is reflected in Section 3.3 for technical parts, and the transfer experiments in Section 4. We made it more clear in the abstract.
>
> Q6: Redundant paragraph before Section 3.
> A6: We removed it.
>
> Q7: Typos.
> A7: Thank you for pointing out. We revised the paper carefully.
>
> Q8: What’s the problem of S2S-CL? Increasing input number n and output number m?
> A8: S2S-CL is Sequence-to-Sequence Continual Learning problem. It is not about increasing input number n and output number m, but about increasing input and output vocabulary sizes. (Section 3.1)
>
> Q9: What does "COMPOSITIONALITY’’ mean in Section 3.2? What’s the relationship between the last two equations of Page 4?
> A9: In Section 3.2, compositionality means the language property that syntax and semantics can be separated, and the output syntactic information depends only the input syntactic information, and (given output syntactic information,) the semantic output information depends only on the input semantic information. Please refer to (Li, 2019) for more details. The last two equations (Eq. 2 and Eq. 3) on Page 4 are probabilistic interpretation of the compositionality property.
>
> Q10: Simplifications in the first Equation of Page 5?
> A10: This simplification is valid because we design the model in the way that for each output label, Y^f tells which x^p corresponds to the label, and X^p tells the value of x^p, so that this label can be inferred without knowing other labels. (Eq. 4)
>
> Q11: Notation "{0,1}^{Uxn}’’.
> A11: We borrow the notation from (Li, 2019). We now do not use the notation, but explain it in the text.
>
> Q12: Entropy regularization with L2 norm.
> A12: Adding noise and using L2 regularization together reduce channel capacity (amount of information it can contain) in each representation and thus entropy for the representations. Please see Section 2 in (Li et al 2019) for more information.
>
> Q13: What are the detailed settings of the demonstrated experiments?
> A13: This is explained in the first paragraph of Section 4 and Appendix A provides more details. Please see Table 2 for simple examples. In summary, the experiments include two stages. The first stage is a standard process in which we train a model with combinations of multiple words in various sentence structures. In each continual stage, we add a new input word and corresponding new output symbol. The training dataset contains only one sample, whose input is a sentence with the new word, and output is a sequence with the new symbol. For each continual stage, we can only use the data in that stage, and have no access to data in previous or future stages.

---

### Author Response · Authors · 2019-11-15
**Response to all reviews**

Thank you for reviews. We summarized some updates based on the suggestions.
- Added flowchart in Figure 1.
- Added standard deviations in Figure 3.
- Revised the paper to improve writing and language.
- Made the overall logic more clear.
- Removed some redundant texts.
- Corrected typos and confusing notations.

---

### Decision · Program_Chairs · 2019-12-19

**Decision:**

Accept (Poster)

**Comment:**

The paper addresses the task of continual learning in NLP for seq2seq style tasks. The key idea of the proposed method is to enable the network to represent syntactic and semantic knowledge separately, which allows the neural network to leverage compositionality for knowledge transfer and also solves the problem of catastrophic forgetting. The paper has been improved substantially after the reviewers' comments and also obtains good results on benchmark tasks. The only concern is that the evaluation is on artificial datasets. In future, the authors should try to include more evaluation on real datasets (however, this is also limited by availability of such datasets). As of now, I'm recommending an Acceptance.